# Towards LoRaWAN without Data Loss: Studying the Performance of Different Channel Access Approaches

**DOI:** 10.3390/s22020691

**Published:** 2022-01-17

**Authors:** Frank Loh, Noah Mehling, Tobias Hoßfeld

**Affiliations:** Institute of Computer Science, University of Würzburg, 97074 Würzburg, Germany; noah.mehling@informatik.uni-wuerzburg.de (N.M.); tobias.hossfeld@uni-wuerzburg.de (T.H.)

**Keywords:** LoRaWAN, IoT, channel management, scheduling, collision

## Abstract

The Long Range Wide Area Network (LoRaWAN) is one of the fastest growing Internet of Things (IoT) access protocols. It operates in the license free 868 MHz band and gives everyone the possibility to create their own small sensor networks. The drawback of this technology is often unscheduled or random channel access, which leads to message collisions and potential data loss. For that reason, recent literature studies alternative approaches for LoRaWAN channel access. In this work, state-of-the-art random channel access is compared with alternative approaches from the literature by means of collision probability. Furthermore, a time scheduled channel access methodology is presented to completely avoid collisions in LoRaWAN. For this approach, an exhaustive simulation study was conducted and the performance was evaluated with random access cross-traffic. In a general theoretical analysis the limits of the time scheduled approach are discussed to comply with duty cycle regulations in LoRaWAN.

## 1. Introduction

The introduction of smart solutions in most applications of the everyday life is one of the fastest growing and most dynamic technology trends nowadays. Although requirements for, among others, Industry 4.0 solutions, traffic management systems in Smart Cities, or simple weather forecasts are completely different from a technology perspective, they have one in common: the requirement for any form of data acquisition.

One solution to create large scale sensor networks are widespread 5G networks. The drawback is an expensive deployment, although often only a minimum of the possibilities a 5G network can provide is required. A different approach is possible with the so-called Low Power Wide Area Networks (LPWANs) with Long Range Wide Area Networks (LoRaWANs) as one of the most prominent representatives. LoRaWANs promise economically priced infrastructure, long battery life times for devices, and large transmission distances at the cost of low data rates and a less reliable transmission. Despite these drawbacks, the popularity of LPWANs increased in recent years. Between 2018 and 2019 an increase of 110% is recorded [1], for LoRaWAN in particular, 700 million connected devices are expected by 2023 [2]. For that reason, it is important to improve the reliability as the major drawback of the technology to foster deployment.

The main reason for the low reliability in LoRaWAN is random channel access with high potential for message collisions, in the worst case, data loss. However, alternatives are already investigated with slotted ALOHA, listen before talk, and time scheduled channel access. However, general guidelines and comprehensive statements for the usage, especially in coexistence with other channel access approaches are not, or only partially [3], taken into consideration so far in the literature. In addition, most related work does not take the duty cycle regulations of LoRaWAN into consideration that apply also for the gateway to avoid an overload in the system by signaling and synchronization messages.

For that reason, a comprehensive simulation study was performed in this work to evaluate the performance of different channel access methodologies in more realistic conditions with different, especially randomly chosen, message sizes, and spreading factors. This leads us to the following research questions for this work:Is it possible to schedule the messages in a LoRaWAN channel to avoid collisions completely despite the device and LoRaWAN specific challenges such as device clock drifts and gateway duty cycle?How does random access cross-traffic, that is not avoidable in a free to use access technology, influence such a scheduled access strategy?

Based on these research questions, the contribution of this work is threefold. First, general guidelines are presented to use each channel access strategy individually. In addition, the collision probability as a quality metric is quantified by simulations. Second, a time scheduled channel access approach is presented and a general relationship between the possible number of messages that can be transmitted within a specific time frame, the device behavior by means of their clock inaccuracy, and the re-synchronization potential is presented. This shows the theoretically possible limit in LoRaWAN to avoid collisions completely without violating the duty cycle regulations. The third contribution is an exhaustive simulation to study the performance of the time scheduled approach in coexistence with random access cross-traffic. This study demonstrates the value of the scheduled approach if cross-traffic can be kept low or additional message recovery is possible.

In the remainder of this work, general background information and related literature is given in Section 2 and Section 3, respectively. The used methodology and the simulation approach is introduced in Section 4. In Section 5 results for different scenarios are discussed and Section 6 concludes.

## 2. Background

This section summarizes fundamental background information briefly that is required to understand this work. More details are given in the Semtech datasheet [4]. First, important technical information to understand LoRa messages and the transmission with LoRa in LoRaWAN is given. Afterwards, details about LoRaWAN channel access are presented. An overview of important notations that are used throughout this work is given in Table 1.

### 2.1. LoRa and LoRaWAN

LoRa is the physical layer LPWAN modulation technique based on the chirp spread spectrum technology [5] used in LoRaWAN. The most important parameter to study LoRa transmissions and especially channel access is the time on air (ToA). It determines the duration a LoRa messages occupies a single transmission channel and is limited for every device in the network by the duty cycle. The duty cycle is a transmission time limitation agreement in LoRaWAN channels by all devices, and in particular also by the gateway, to not exceed channel occupancy limitations and leave free channel time for other devices in the network. Duty cycle limits are 0.1%, 1.0%, or 10% occupancy time per hour, dependent on the used frequency band. Furthermore, the ToA is influenced by several adjustable parameters. The most important ones are the bandwidth (BW), the spreading factor (SF), and the total message size. A summary of all LoRa related parameters with the respective values used in this work is given in Table 2. More details about the LoRa modulation and message structure is available in [6]. Since the 868 MHz frequency band (EU868) is typically applied in Europe with 125 kHz channel width, this is selected in this work.

#### 2.1.1. Spreading Factor

The *SF* in the range from SF 7 to SF 12 in LoRa determines the number of raw bits a symbol carries. For example, one symbol transmitted with SF 12 can be mapped to 12 bit. The symbol duration is achieved based on the *SF* and the *BW* with
(1)$$T_{s}=\frac{2^{\mathrm{SF}}}{\mathrm{BW}}.$$

Thus, the usage of a higher *SF* results in a longer symbol duration but the signal can also be transmitted across longer distances and is more robust against interference.

#### 2.1.2. LoRa Message

A message in LoRa contains a preamble, 4.25 symbols for synchronization, an optional header, and the actual LoRa payload. In this work, a preamble length of eight symbols is used according to the default value [5]. The required number of symbols for the LoRa payload is achieved by
(2)$$n_{\mathrm{payload}}=8+max\left(\left\lceil (n_{\mathrm{packet}})\right\rceil \;\cdot \;\left(\mathrm{CR}+4\right),0\right)$$
with
(3)$$n_{\mathrm{packet}}=\frac{\left(8\mathrm{PL}-4\mathrm{SF}+28+16\mathrm{CRC}-20\mathrm{IH}\right)}{4(\mathrm{SF}-2\mathrm{DE})}.$$

Details about each parameter are given in Table 2. The number of symbols for a single LoRa message is
(4)$$n_{\mathrm{message}}=P+4.25+n_{\mathrm{payload}}.$$

With the symbol duration $T_{s}$, the ToA $T_{m}$ of a LoRa message can be calculated with
(5)$$T_{m}=T_{s}\;\cdot \;n_{\mathrm{message}}.$$

### 2.2. LoRaWAN Channel Access

Channel access planning in LoRaWAN is a complex task. Challenging factors are, among others, different channel access approaches, limited synchronization possibilities due to duty cycle limitations for class A devices and the gateway, and strict requirements at the end devices to save battery. Furthermore, the free usage possibility of the frequency bands can lead to potential cross-traffic. Please note: since class A devices in LoRaWAN have most limitations in availability for re-synchronization by returning frequently in deep-sleep to save energy, these devices are used for the analysis in this work. However, the suggested approaches also work for class B and class C devices, respectively.

Currently, a random access approach similar to the pure ALOHA protocol is used in LoRaWAN. However with pure ALOHA, the maximal channel utilization is only 18.4% [7] and dealing with collisions is an important task in current LoRaWAN. One solution is to transmit acknowledgments if messages are received correctly. With this solution additional messages must be created, the transmission duty cycle of the gateway is charged, and additional collisions are possible. For that reason, different alternative channel access schemes have been introduced in recent years to cope with this challenge.

#### 2.2.1. Slotted ALOHA

One potential improvement is slotted ALOHA. It divides the frequency channels into time slots and allows channel access by slot allocation. End devices have to conform to these time slots and only initiate transmissions at the beginning of a slot. As a result, collisions can only occur if two or more devices transmit in the same time slot instead of being at a constant risk of interference from other devices such as in pure ALOHA. This setup theoretically allows slotted ALOHA to reduce the number of collisions as well as the vulnerable time by 50% when compared to pure ALOHA [8]. Furthermore, the maximum channel utilization is increased to 36.8% [7]; however, accurate timing information and re-synchronization is required to keep devices aligned to the time slots and clock inaccuracies resulting in clock drifts must be taken into consideration. This is dealt with by using an appropriate slot length and additional guard times to compensate slight inaccuracies.

#### 2.2.2. Listen before Talk

Another possible channel access strategy is listen before talk. There, the devices listen on the channel before attempting a transmission and only transmit if the channel is free. The benefit of this strategy is the avoidance of overhead by synchronization or additional channel access control. However, if a device is attempting to transmit data and the channel is occupied, the transmission is delayed by a specific form of back-off delay. According to the literature, there are different approaches to determine this back-off delay duration. One idea is to create back-off slots of a specific duration and delay the transmission for a specific number of such slots [3]. Furthermore, the back-off delay can be combined with frequency hopping [9]. In total, the interference between devices in a network with many end devices trying to send at the same time can be reduced [10]. However, the possible performance listen before talk can achieve is limited by the hidden node problem. When all nodes in a network are hidden from each other, the performance of listen before talk is reduced to the same level as pure ALOHA [10]; in reality, many nodes can be hidden from others due to the long range of LoRa connections [10].

## 3. Related Work

This section summarizes background literature for LoRaWAN and related works regarding LoRaWAN channel access. Furthermore, the difference to this work is highlighted and other simulation-based approaches are presented.

One pioneer work to present a general overview of the limits in LoRaWAN is given by Adelantado et al. [5]. The authors study important LoRa parameters such as SF or payload size and highlight limitations in LoRaWAN. Since then, many works are published that rely on different strategies to investigate the performance of LoRaWAN. For example large scale measurement studies in real environments are tackled by many works, e.g., [11,12,13]. In addition, more specific impact factors on LoRaWAN quality such as multiple gateways [14], gateway placement [15], performance in critical environments [16], temperature [17], or performance comparison between indoor and outdoor deployment [18] are tackled; however, large scale measurement studies are time consuming and expensive. For that reason, other authors deploy simulation-based approaches for, among others, scalability in urban environments [19] or SF influence on message collision behavior [20]. Other authors simulate LoRaWAN with common simulation tools such as ns-3, summarized by da Silva in [21], with Matlab [22], or other approaches [23]. Furthermore, in [24] a software to evaluate own LoRaWANs by means of collision probability is presented. In addition, general statements regarding LoRaWAN performance are made. For example cell capacity is studied in [25].

The robustness and possibility to decode signals is studied in different works for different IoT technologies (e.g., [26,27,28]). This research direction is important, in particular for LoRa, because of the usage of the chirp spread spectrum and different SFs. In [29], the authors study the SF orthogonality and the influence on the overall throughput. In [30], it is proven that the exact collision position in one LoRa message and the SF is important to decide whether a packet can be recovered after a collision and [31] shows that the usage of this fact can decrease loss in LoRaWAN. Recently, Shahid showed that for a specific percentage of LoRa messages, it is possible to decode multiple messages transmitted at the same time with the same SF [32]; however, instead of decoding colliding messages it is possible to avoid simultaneous transmissions by intelligent channel access approaches.

Bankov studied the limits of LoRaWAN channel access first in 2016 [6]. Since then, especially slotted ALOHA, listen before talk, and scheduled MAC-based approaches are presented [10]. For slotted ALOHA, among others, a methodology to increase single channel capacity is proposed [33], different back-off schemes to avoid collisions are presented [34], or aggregated acknowledgments are proposed to improve scalability and reliability for scheduled transmissions [35]. Garrido studies scheduling in LoRaWAN and presents a real-world implementation recently in [36]. Furthermore, CSMA-based approaches are discussed as listen before talk solutions [37,38,39]. In addition, sole listen before talk approaches are studied and compared to pure ALOHA in literature [9,40], while explicitly the coexistence of listen before talk and ALOHA is studied in [3]; however, often neither the created overhead nor the duty cycle usage for the devices or the hidden node problem is studied in literature.

## 4. Methodology

This work deals with a thorough simulation study for channel access in LoRaWAN. For that reason, details about each channel access methodology discussed in this work are presented in this section. Furthermore, parameter settings that are important for the evaluation of the presented approaches are determined. At the end of this section, the further studied scenarios are described in detail; however, some preliminary considerations are summarized initially.

### 4.1. Preliminary Considerations

The variation of different LoRa-specific transmission parameters, the transmitted payload, and the SF lead to a different ToAs for LoRa messages according to Equations (Equation 2)–(Equation 5); however, for the channel access methodology and interference detection due to messages transmitted simultaneously within one LoRaWAN channel, only different channel occupancy durations, and thus the ToA is relevant. For that reason, channel access studies are focused on the ToA only in this work. Additional parameters such as the SF are only taken into consideration for additional message recovery mechanisms in case of simultaneous channel access according to [30].

Since 51 B is the largest possible payload for messages transmitted with SF 12 in LoRaWAN, this is set as payload limit in this work. All possible message ToAs for a payload between 1 B and 51 B are calculated and presented in Figure 1. The ToA in seconds is shown on the *y*-axis and the used payload in bytes on the *x*-axis. The colors present the different SFs. The total ToA range is from 0.029 s for 1 B transmitted with SF 7 up to 3.023 s for 51 B transmitted with SF 12 and the parameter settings of Table 2. In addition, it is visible that the same ToA can be achieved by different payload and SF combinations.

Furthermore, LoRa messages can be sent by a multitude of different devices in a LoRaWAN or multiple messages can be sent by a single device; however, to determine the channel access behavior and especially the collision behavior, the number of devices in the network is not important but only the total number of sent messages. For that reason, only the number of messages transmitted in a specific time span is considered for the channel access and the evaluations. The transmitting devices are only taken into consideration for specific clock drifts or for further location specific considerations in the course of this work.

### 4.2. Channel Access Approaches

This section presents alternatives to random channel access that are studied in the course of this work. The goal in this section is to limit the total parameter range. Furthermore, the usability of slotted ALOHA for LoRaWAN is discussed and a time scheduled approach is presented as an alternative approach with dedicated transmission slots. Afterwards, the methodology for listen before talk as an alternative to random access is introduced and important parameters are studied. At the end of this section, additional possible message recovery mechanisms in case of collisions are addressed.

#### 4.2.1. Slotted ALOHA

Channel access with slotted ALOHA is optimal when all slots are used, each message has a slot with its individual length without wasting channel resources and no message drifts out of its individual slot; however, several challenges prevent optimal slotted ALOHA in LoRaWAN. In particular, channel occupancy times for different messages vary between 0.029 s and 3.023 s as shown in Figure 1 in real LoRaWAN deployments. Long slot lengths equal to the maximal ToA waste much resources when small messages are transmitted and small slot lengths can make it impossible to transmit longer messages. For that reason, a reasonable slot length selection in real LoRaWAN deployments with different payloads and SFs and thus, different ToAs is challenging.

If a random payload and SF is selected, we see in a pre-study that slotted ALOHA with a slot length equal to the maximal ToA of 3.023 s achieves worse results with regard to the collision probability compared to the state-of-the-art random access solution (see Appendix A). For that reason, we suggest a time scheduled approach with the same challenges but several improvements as better alternative to slotted ALOHA.

#### 4.2.2. Time Scheduled

In contrast to slotted ALOHA where the device is using the next free slot depending on the transmission request time, each device in a LoRaWAN must register for a transmission slot at the gateway in a time scheduled (Scheduled) approach. After the slot is assigned, it is committed to use only this specific slot. With this idea, Scheduled can avoid collisions, and thus data loss, completely if each device keeps to its slots.

However, random clock drifts or delays as result of limited computational resources occur [33] in reality and prevent perfect Scheduled. Furthermore, since time synchronization possibilities between devices and the gateway are limited because of duty cycles regulations in LoRaWAN, the determination of perfect individual slot lengths and guard times for each message is a challenging task. In the following, this is achieved by further analysis.

##### Clock Drifts

Clock drifts occur due to the nature of many oscillator crystals used in lower cost devices. These oscillators produce an uncertainty of timing if running too slow or too fast. This uncertainty can be expressed as a deviation from the nominal frequency in a parts per million (ppm) unit. According to literature, common drifts are 0.5 ppm, 2 ppm–10 ppm, or 10 ppm–100 ppm [41]. In total, a clock drift of 1 ppm is equal to a drift of 3.6 ms per hour. In addition, no linear drift is achieved in reality due to, among others, different temperature [41].

Clocks with very little drift are comparable expensive and not applicable for a large scale LoRaWAN deployment. For that reason, drifts smaller than 2 ppm are not taken into consideration in this work. In addition, too large drifts result in large timing uncertainties per hour. This leads to the necessity of large guard times, slot lengths, or frequent re-synchronization with additional packets and overhead to avoid collisions by messages drifting out of their time slots. Furthermore, frequent re-synchronization is not advisable since it limits or exceeds the gateway duty cycle. Thus, if messages are sent by devices with a large clock drift, additional self-calibration or correction approaches are suggested [42]; however, this is not the focus of this work. For that reason, acceptable clock drifts are outworked for the Scheduled approach in LoRaWAN in the following. Furthermore, a slot length study was performed to determine a reasonable setting.

##### Slot Length

It must be guaranteed in LoRaWAN that no messages are transmitted at the same time to avoid collisions. Thus, the maximal possible number of messages that can be transmitted in a predefined time span can be determined by the slot length. Messages larger than their slots create systematic collisions with their slot neighbors for each transmission. If no recovery mechanism is possible or available, these systematic collisions can prevent individual devices from being able to successfully transmit any further message to the gateway in specific time slots. Thus, this behavior can break the complete system. No collisions with Scheduled for LoRaWAN without any cross-traffic can be guaranteed with a minimal slot length of $L_{s}$ according to
(6)$$L_{s}=\max\left(T_{m}\right)+\max\left(T_{\mathrm{sync}}\right)+\max\left(T_{d}\right)+r$$
with $\max(T_{m})$ as maximal ToA among all messages in the system, max($T_{\mathrm{sync}}$) as ToA for the largest re-synchronization message, and $\max(T_{d})$ as maximal clock drift of a single device. Furthermore, a random variable *r* is added for potential randomness in the clock drift. Please note that the possibility of receiving re-synchronization messages at the end device immediately after the transmission is finished must be guaranteed for a minimal slot length of $L_{s}$. Otherwise, an additional constant *c* must be added for the duration between message transmission end and reception window opening. Furthermore, additional delays by among others transmission and processing delays are neglected; however, a slot length of $L_{s}$ requires re-synchronization after each transmission for all messages with $\max(T_{m})$ as maximal ToA and large clock drifts. To avoid this, and especially avoid re-synchronization of specific messages after each transmission, initially twice the maximal clock drift of $T_{d}$ is added. This leads to a minimal slot length of $L_{s}^{\prime }$ for further studies with
(7)$$L_{s}^{\prime }=\max\left(T_{m}\right)+\max\left(T_{\mathrm{sync}}\right)+2\;\cdot \;\max\left(T_{d}\right)+r$$

Furthermore, $L_{s}^{\prime }$ as slot length also guarantees no collision if the clocks of different devices drift in opposite directions and a guard time in positive and negative direction from the slot is added. Please note that this scenario was not further investigated since it is very unlikely for similar climatic conditions [43].

In LoRaWAN, $\max(T_{m})$ is equal to 3.023 s for 51 B payload transmitted with SF 12 and max($T_{\mathrm{sync}}$) is equal to the minimal ToA required to transmit data with one specific SF. Since higher SFs are more robust against interference [30], SF 12 is chosen to transmit the re-synchronization message for the parameter study in this work. Thus, max($T_{\mathrm{sync}}$) is equal to 0.926 s to transmit 1 B up to 6 B as re-synchronization message.

In addition, different maximal clock drifts are feasible; therefore, it is assumed that each device transmits once an hour. Thus, a drift guard time for the maximum clock drift $\max(T_{d})$ up to 100 ppm or 360 ms per hour is applied initially. Please note that other transmission rates for the devices are also practical. In reality, the $\max(T_{d})$ parameter must be adjusted according to the largest clock drift between two transmissions; however, it is advisable to not use Scheduled for too large drifts. The transmission randomness variable *r* is set to 10%. Thus, the tested slot length adds up to
(8)$$L_{s}=3.023\;s+0.926\;s+2\times 0.36\;s+0.036\;s=4.705\;s.$$

For one LoRaWAN channel, 765 messages can be transmitted per hour in 765 slots with perfect synchronization without cross-traffic; however, the re-synchronization is not taken into consideration for this number.

#### 4.2.3. Re-Synchronization Investigation

The gateway duty cycle limits the transmission time of the gateway to 0.1%, 1.0%, or 10% per hour, dependent on the selected frequency band. Thus, it also limits the number of messages a gateway can transmit in a specific time interval. If the number of devices transmitting messages in one channel is increasing or the clock drifts of the devices are larger, more re-synchronization messages are required. A general limit for the number of possible re-synchronization messages per hour based on the duty cycle can be expressed by the following equation:(9)$$n\;\cdot \;T_{\mathrm{sync}}\;\cdot \;P\left(\mathrm{sync}\right)\leq d\;\cdot \;3600$$
with *n* as number of messages per hour, $T_{\mathrm{sync}}$ as message re-synchronization ToA, P(sync) as re-synchronization probability per message, and *d* as duty cycle. Since $T_{\mathrm{sync}}$ is fixed by the SF and the duty cycle by the LoRaWAN regulations, it is only possible to vary the number of transmitted messages per hour and the re-synchronization probability.

To analyze P(sync) in detail, a differentiation between messages per hour and number of transmitting devices per hour must be made since clock drifts are achieved per device and not per message; however, in real IoT systems smaller and larger transmission rates than once an hour are practical [44]. For that reason, it is assumed that each device transmits messages once an hour in the following. For different rates the clock drift values between two transmissions must be adapted accordingly while the calculation keeps the same.

For a fixed maximal message ToA, the message limit per hour can then be determined dependent based on the SF, and thus the ToA of the re-synchronization message according to the tolerated P(sync), shown in Figure 2. A maximal duty cycle for the gateway of 1% is used according to most frequency bands. The tolerated P(sync) drops drastically, especially if large SFs are used for the re-synchronization messages and many messages per hour are sent. It is still possible to re-synchronize all device clocks for example for 500 messages per hour if the re-synchronization message is sent with SF 7 and 96.79% for SF 8. Only 7.76% of all device clocks can be re-synchronized with SF 12 without exceeding the duty cycle. For that reason, the number of devices transmitting messages and the re-synchronization probability P(sync) must be studied in detail.

#### 4.2.4. Re-Synchronization Relationship

The re-synchronization probability is affected by two factors: the clock drift of the transmitting device $T_{d}$ and the drift limit $T_{l}$ before a re-synchronization is performed. This relationship can be denoted as follows:$$\begin{array}{cc} T_{d}>T_{l}\;⟹ & \;P(\mathrm{sync})=1\\ \frac{T_{l}}{2}<T_{d}\leq T_{l}\;⟹ & \;P(\mathrm{sync})=0.5\\ \vdots \;\; & \;\vdots \\ \frac{T_{l}}{k}<T_{d}\leq \frac{T_{l}}{k-1}\;⟹ & \;P\left(\mathrm{sync}\right)=\frac{1}{k} \end{array}$$

If the clock drift of the transmitting device $T_{d}$ is larger than the drift limit $T_{l}$, a re-synchronization message must be sent after each transmission. This leads to P(sync)=1. Each second message must be re-synchronized if $T_{l}/2<T_{d}\leq T_{l}$, etc. This leads to
(10)$$T_{d}\leq \frac{T_{l}}{k}\;⟹\;P\left(\mathrm{sync}\right)<\frac{1}{k}$$
which describes a general relationship between the clock drift of the transmitting device $T_{d}$, the total guard time before a re-synchronization message is sent $T_{l}$, and the re-synchronization probability per message P(sync); however, a larger total slot length has the same effect as large guard times. For that reason, the terminology slot length is used in the following, which sums up the actual slot length and the guard time.

Equation (Equation 10) is validated by a simulation with 765 devices transmitting once per hour and the slot length of Equation (Equation 8). Each clock drift from 2 ppm to 100 ppm in 0.1 ppm steps is simulated for 200 h. The clock drift per hour and the P(sync) is determined. A drift randomness of up to 10% is added randomly.

Figure 3 shows the result with P(sync) on the *y*-axis and the selected drift per hour on the *x*-axis. The black line shows the average simulation result for each clock drift, the orange line the model of Equation (Equation 10) and the dashed brown line a 10% deviation from the model according to the added clock drift randomness. In general, the simulation result describes the model well and the 10% randomness can be approximated with a 10% deviation from the model. For that reason, this result is used to determine the possible number of devices transmitting messages in one hour by means of P(sync) and the clock drift per hour $T_{d}$ in the following. The limiting number of messages per hour *n* can be denoted as
(11)$$n<\left(\max\left(T_{m}\right)+\max\left(T_{\mathrm{sync}}\right)+\frac{2\;\cdot \;\max(T_{d})}{P(\mathrm{sync})}+r\right)d\;\cdot \;3600.$$
In the equation the duty cycle per hour d·3600 from Equation (Equation 9) is multiplied with the possible minimal slot length $L_{s}^{\prime }$ from Equation (Equation 7) depending on the re-synchronization probability P(sync). P(sync) is only dependent on the time drift $\max(T_{d})$, as shown in Equation (Equation 10).

Thus, the limiting number of messages per hour *n* can be determined dependent on the maximal clock drift $T_{d}$ where P(sync) does not exceed the message limit. The result is presented in Figure 4. The figure shows the maximal clock drift $T_{d}$ on the *y*-axis and the possible number of messages per hour on the *x*-axis. The different colors depict the used SF for the re-synchronization message with darker colors for lower SFs. For example for a clock drift of 100 ppm, 430 devices can send with SF 12 within one hour or 925 with SF 7. If the clock drift is only 50 ppm, 540 devices are possible with SF 12 and 1027 with SF 7. Furthermore, since in reality not all messages transmit with $\max(T_{m})$, additional gateway duty cycle is left in real deployments for other purposes such as acknowledgments or device updates. Thus, we can answer our first research question with *yes*, *if the number of messages is chosen according to Equation (Equation 11)*, *we can avoid collisions completely*.

#### 4.2.5. Listen before Talk

Before a device transmits a message with listen before talk, it listens to the channel whether it is already occupied. If it is free, the transmission is started and otherwise the message is delayed according to a predefined back-off strategy. Thus, no additional synchronization is required. However, in a real world deployment, each device has a geographic location where it transmits from and a specific distance to the gateway. Thus, only other devices in close distance to the transmitting device can hear potential messages. If another device is outside this transmission radius but in the same cell and thus, transmitting to the same gateway, collisions can occur because of the hidden node problem. In the following, first a possible back-off strategy is studied if a device attempts to transmit but finds an occupied channel. Afterwards, a location and transmission distance calculation is presented.

##### Back-Off Strategy

The back-off strategy for listen before talk determines the duration a message is delayed after an unsuccessful transmission attempt when the channel was occupied. To study the performance of the back-off strategy, only devices in the transmission radius of the sending device are of relevance. For that reason, for this study, transmitting devices are assumed to be heard by all others. To determine the used back-off strategy, fixed and random back-off delays were compared. In this study, 100 fix back-off delays between the minimal ToA of 0.029 s and the maximal ToA of 3.023 s were simulated and compared to a random back-off in the same time range. For each parameter setting, 20 simulation reruns for 200, 500, and 1000 messages per hour were performed. A similar result is achieved for all message numbers while clearer differences are achieved with more messages per hour.

Thus, the result of the simulation for 1000 messages per hour is presented in Figure 5. The left *y*-axis shows the average number of delays for a single message before the transmission attempt is successful. The *x*-axis shows the set back-off delay in seconds for fixed delays. The brown line shows the result for this study. In comparison, the black solid line at 0.29 s shows the average number of delays if the back-off is randomly chosen.

It is visible that a fixed back-off delay shows worse results for back-offs below 1.0 s. In addition, many attempts are required before the transmission is successful if a single back-off delay is small. For a larger back-off delay, the result is comparable to the random delay and slightly better for a delay larger than 1.25 s. This behavior is explained as follows: if the back-off delay is very small, the probability is high that one message $M_{1}$ with a transmission conflict with message $M_{2}$ is delayed several times before the transmission start time of $M_{1}$ is delayed enough to avoid a collision with the transmission end of $M_{2}$ anymore. In contrast, for a larger back-off delay, this probability decreases; however, with a longer back-off delay, messages might be delayed unnecessarily long. This is highlighted by the right *y*-axis and the yellow line presenting the total average delay per message in relation to the back-off delay in seconds. The dashed black line at 0.45 s shows the total average delay time per message with the random back-off selection. The result shows an increasing total average delay with larger back-off delay. Thus, although the message is delayed less often for a larger back-off delay, the total delay is larger.

For that reason, a trade-off between small back-offs with many delays and large back-offs with a large total delay must be determined. Since the random back-off approach shows promising results and the fixed back-off approach shows good results between 0.4 s and 1.75 s, a random back-off in this range is simulated. The goal is to avoid multiple back-offs per message and unnecessary long back-offs. In this study, the average number of delays is 0.32, for the mean back-off scenario with the full back-off time range between 0.029 s and 3.023 s it is 0.29. The mean delay when using a back-off delay randomly between 0.4 s and 1.75 s is 0.34 s, for the full range back-off it is 0.45 s. A single message is delayed for a maximum of 16 times for the random approaches and 277 times when the smallest delay of 0.029 s is used. For that reason, in the course of this work, the back-off delay is selected randomly between 0.4 s and 1.75 s. Please note that slightly better results might be possible with further studies.

##### Location and Transmission Distance

In real network scenarios, devices that send messages are located in geographically distributed locations; thus, not each device can hear all messages in the network. To emulate this behavior, all devices and the gateway are allocated an *x* and *y* coordinate with the following strategy: first, the maximal transmission distance $d_{M}$ for SF 12 is calculated with the Hata model according to [45]. The gateway and the device height is 3 m. Please note that any other model, parameter setting, or maximal distance is reasonable too and will not change the general statement. The device height and the gateway height are set to the same values to guarantee bidirectional reachability. The resulting maximal transmission distances are summarized in Table 3 with the tolerated RSSI limits from [46] and a commonly used device transmission power of 8 dBm.

Afterwards, the coordinates $x=y=d_{M}$ are allocated to the gateway and all devices are randomly placed around the gateway within a radius of $d_{M}$. Please note that other device placement strategies are also applicable. In this way, it is guaranteed that each device can transmit to the gateway in this simulation but not every message is heard by any device. In a last step, the distance to the gateway is calculated for each device and the minimal possible SF to access the gateway is assigned to the device and all messages transmitted by it.

#### 4.2.6. Message Recovery

Without message recovery approaches, colliding messages are lost. This is especially crucial for re-synchronization messages for Scheduled; however, according to the literature, not all colliding messages are lost [30,32]. Thus the simulation in this work is extended by a recovery approach where it is assumed that messages transmitted with higher SFs always dominate messages transmitted with lower SFs and are thus transmitted correctly [30]. Messages with the smaller SF are lost. Furthermore, the collision position in the message is of high relevance for potentially lost messages [30]; however, informing the transmitting device to re-transmit an error prone message requires additional overhead. Thus, this is not studied in this work and left as future work. Furthermore, additional potential to decrease message loss is the usage of the quasi orthogonality of different SFs; however, according to literature collisions still occur when different SFs are used [47]. Thus, this is not applied.

### 4.3. Simulation Methodology

The goal of this simulation study was to determine the performance of the presented channel access approaches and study the Scheduled approach in coexistence with cross-traffic. To understand the general idea, this section introduces the simulation methodology, and assumptions for the simulation.

#### 4.3.1. General Simulation Idea

The general simulation idea is based on five steps summarized in the following (the simulation for data generation and evaluation scripts are available at Github https://github.com/lsinfo3/lora_without_collision_sensors, (accessed on 16 December 2021).

**Step** **0:**Before the simulation starts, channel access specific parameters are created and added to the respective messages. In addition, the simulation duration is determined and the simulation scenario is specified. During the complete simulation, one simulation iteration can be specified as a specific time frame. This time frame, set to one hour in this work, is selected and simulated at once, which makes the simulation very fast and lightweight in contrast to reference simulations from related work. Please note that different time frames work accordingly while duty cycle constraints must be recalculated then.**Step** **1:**In the first simulation step, all transmission start timestamps are created and by adding the randomly generated ToA, the transmission end timestamps are calculated.**Step** **2:**For the complete time frame, it is determined whether transmission intervals overlap. For all overlapping intervals it is determined whether the messages collide or if any channel access specific or message recovery specific methodology prevents the collision.**Step** **3:**The collision probability and other quality metrics for all messages transmitted within this simulation iteration are gathered within this step.**Step** **4:**Additional actions are performed dependent on the channel access, messages with a transmission end timestamp larger than 3600 s are considered for the next simulation iteration and the next iteration is started with step 1.

#### 4.3.2. Start Timestamp Calculation

The transmission start time calculation for all messages in each iteration is different depending on the channel access type. For random access and listen before talk, random transmission start times are calculated for each simulation iteration between 0 s and 3600 s with six digits of precision. This is valid for two reasons: first, the individual behavior of a single message is not studied and is not important for the general statement. Second, if the number of sent messages is sufficiently large, each message can be handled in an independent way according to [44]. Please note that random traffic generation is the most challenging one for scheduled access. Thus this is chosen for comparison to other approaches in this work.

In contrast, the one hour time frame of one simulation iteration is divided in *n* individual slots dependent on the number of messages *n* for the Scheduled approach. Then, the transmission start timestamp for each message is set to one slot start while the transmission end timestamp is calculated by adding the ToA to the start timestamp. Furthermore, a random clock drift between 0 s and the maximal clock drift $T_{d}$ dependent on the drift limit is added at the beginning of the simulation to avoid an unrealistic start condition with all devices starting with a clock drift of zero. Initial studies show that with this approach no transient phase in the simulation is detected for Scheduled and thus, no simulation results must be discarded. Further parameters are only relevant for specific channel access strategies and thus, mentioned in the respective section hereafter.

#### 4.3.3. Message Collision and Message Recovery

Two consecutively sent LoRa messages $M_{i}$ and $M_{j}$ with transmission start timestamps $t_{s_{i}}<t_{s_{j}}$ collide when the transmission intervals of $M_{i}$ and $M_{j}$ overlap since the transmission end timestamp $t_{e_{i}}$ of $M_{i}$ is larger than $t_{s_{j}}$. After a collision is detected, either both messages are lost or if messages can be recovered because of higher SF, the message with the larger SF is not affected by the collision and the other one is lost according to Section 4.2.6. If both messages are transmitted with the same SF, both are lost.

#### 4.3.4. Assumptions

Before a scenario definition is possible, several assumption must be made. First, each transmitted message in the network is assumed to reach the gateway. Multi-gateway setups or message losses due to missing gateway reachability are not considered in this work and left as future work. Furthermore, a message can be transmitted from device to gateway and vise versa across the same distance. Thus, if a message arrives at the gateway with a specific SF and the gateway replies with the same or a higher SF, it is assumed that the message arrives at the device if no collision occurs. Last, messages transmitted at the same time, independently of the transmission direction (from gateway to device or vise versa) collide.

### 4.4. Scenario Overview

The theoretical observation showed that the Scheduled approach has the potential to avoid collisions in LoRaWAN completely. For that reason, the focus of the scenario study is the analysis of the coexistence of the Scheduled approach with random access or listen before talk to see the behavior in a real network with cross-traffic. Each parameter combination for each simulation runs for 200 h to avoid potential outliers. Furthermore, ten re-configurations were performed for each simulation where slots and clock drifts are newly assigned to the messages and locations are newly calculated for the LBT devices. This should avoid potential errors by randomly chosen well suiting setups. In the following, details about all scenarios are given.

#### 4.4.1. Scenario S1: Scheduled and Random Access

Scenario S1 is defined as a combination of Scheduled and random access traffic. The clock drift limits for Scheduled are varied between 2 ppm and 150 ppm in this simulation. Accordingly, the number of messages per hour transmitted with Scheduled are adapted according to the findings in Section 4.2.4. The studied maximal clock drifts $T_{d}$ and the associated number of messages *n* are summarized in Table 4. The k-variable is introduced in Equation (Equation 10) as reciprocal value of the re-synchronization probability P(sync). For all messages, the payload and the SF is chosen randomly in the respective value range and the clock drift is assigned randomly between 0 ppm and the limit for the specific simulation. Additionally, a 10% randomness in positive and negative direction is added. Scenario S1 has three sub-scenarios, S1.1, S1.2, and S1.3.

**S1.1:** Scheduled is seen as main channel access methodology while random channel access is additional cross-traffic. The number of messages *n* transmitted with Scheduled for specific maximal clock drifts $\left(T_{d}\right)$ is selected according to the values in Table 4. Additionally 1–50% cross-traffic is added and thus, more messages must be handled by the channel. For example for 150 ppm clock drift with 50% random access traffic, 370 Scheduled and 185 random access messages are transmitted. The highest number of messages transmitted with random access for all scenarios is 437 for 50% random access traffic in scenario S1.1. A pre-study shows that this number of cross-traffic does not overload the network alone and the collision probability when simulating random access individually is less then 15%.**S1.2:** The maximum number of devices *n* is set to the numbers listed in Table 4 for specific maximal clock drifts $\left(T_{d}\right)$. If more random access traffic is transmitted, less traffic is sent with the Scheduled approach and vice versa. For example for 50% random access traffic and a clock drift of 150 ppm, 185 random access messages, and 185 Scheduled messages are transmitted.**S1.3:** The number of messages is analogous to S1.2. In contrast, the slot length is adapted towards the number of messages sent with Scheduled. If no random access traffic is in the network, the slots are kept the same, if 50% random access traffic is in the network, only 50% of the original Scheduled traffic is in the network, and thus, the slot lengths are doubled.

For each sub-scenario, in total eleven different drift settings with eleven different random traffic percentages, ten re-configurations of clock drifts and slot allocations, and 200 simulated hours for each combination were performed. This sums up to 26.63 simulated years for each sub-scenario.

#### 4.4.2. Scenario S2: Scheduled and Listen before Talk

In scenario S2, Scheduled and listen before talk is used for channel access since listen before talk improves the collision probability compared to random access. (For a detailed study of random access and listen before talk, we refer here to Appendix B for the interested reader.) Scenario S2 is similar to S1 with three sub-scenarios, S2.1, S2.2, and S2.3 and the same settings as S1. The goal is to determine if listen before talk cross-traffic coexists better with Scheduled compared to random access traffic. The total simulation duration for each sub-scenario is again 26.63 years.

## 5. Evaluation

This section presents evaluation results for the defined scenarios. The main goal of the scenario study is to answer the second research question, whether Scheduled can coexist with random access cross-traffic.

### 5.1. Scenario S1: Scheduled and RA

The percentage of random access traffic and the clock drifts of the devices transmitting with Scheduled are the main influencing factors in scenario S1. The results of this simulation study are presented in the following.

#### 5.1.1. Traffic Study

The collision probability for different random access traffic percentages is shown in Figure 6. The boxplots present the percentage of random access traffic on the *x*-axis and the collision probability on the *y*-axis. One box includes the collision probability results for all possible clock drifts and re-configurations while each combination was simulated for 200 h. Thus, each boxplot contains more than two simulated years.

The different colors show the different sub-scenarios as described in Section 4.4. The collision probability significantly increases with the percentage of random access traffic, in the first glance independently on the sub-scenario. For up to 15% random access traffic, no clear difference between the sub-scenarios is visible. A median collision probability of less than 5% is only achieved for less than 10% random access traffic. Larger differences between the sub-scenarios are achieved for more random access traffic. For 50% random access traffic, S1.1 performs worse than S1.2 and S1.3. S1.2 is similar to S1.3. The largest influencing factor for collision probability is the percentage of random access traffic. In contrast, the total number of messages has a smaller influence. For 50% random access traffic, in S1.1 50% more messages are transmitted than in S1.2 or S1.3; however, the median collision probability is increased by less than 2%.

#### 5.1.2. Clock Drift Study

The influence of clock drifts, independent on the percentage of random access traffic is studied in Figure 7. It shows the collision probability for maximum clock drifts between 2 ppm and 150 ppm. The collision probability is decreasing with increasing clock drifts since in total less messages must be transmitted per hour according to Table 4. The behavior is independent on the sub-scenario. The outliers are a result of the large range of random access percentages together with the Scheduled channel access.

#### 5.1.3. General Result

In general, the percentage of random access traffic is the largest impact factor on the collision probability. Thus, much random access traffic does not coexist in a good way with Scheduled. Furthermore, the total number of transmitted message per hour has a larger impact on the overall collision probability than the clock drift of the devices transmitting data with Scheduled. For that reason, the collision probability is especially high for small clock drifts where it is possible to transmit many messages per hour. Furthermore, a larger variance in collision probability is detected with 0.67% for 1% random access traffic for a clock drift of 2 ppm in contrast to 23.56% for 50%. A collision probability of only 0.33% is achieved for 150 ppm, with 1% random access traffic and 10.78% for 50% respectively. Thus, we can answer our second research question with *Scheduled can coexist well with random access-based cross-traffic if Scheduled is the predominant form of traffic.*

#### 5.1.4. Individual Collision Study

Next, the collision probability is investigated based on the colliding message type. This is shown for S1.1 in Figure 8. The collision probability for random access messages shown by the black boxplots is a lot higher compared to the Scheduled traffic shown by the brown boxplots for all random access traffic percentages. The median collision probability is more than 20% for all random access traffic values and increases significantly starting from 4% random access traffic. The variance of the result is high, especially for few transmitted random access messages. In contrast, Scheduled performs much better for all percentages even if 50% random access messages are sent.

To quantify the quality of the Scheduled channel access in coexistence with random access, not only the collision probability for random access and Scheduled messages is taken into consideration. It is important to know whether Scheduled messages collided with random access message or other Scheduled message. If two Scheduled messages collide, a complete impairment of the general Scheduled access methodology can happen. If the clock of one device drifted that far out of its channel without re-synchronization because of frequent collisions of the re-synchronization messages, two Scheduled messages collide. In the worst case, this collision reoccurs frequently each hour and makes re-synchronization impossible. This is called *systematic collision* in the following and is studied in addition to random access and Scheduled collisions. Furthermore, the collisions of re-synchronization (sync) messages are investigated.

With regard to systematic collisions for scenario S1.1, a maximal collision probability of 0.54% for a single iteration with 50% random access traffic is achieved. Only a single systematic collision is detected for 1% random access traffic for more than 2.5 simulated years. This leads to a systematic collision probability of 0.0003% for 1% random access traffic and a 13-times increase for 50% random access traffic. A significant increase is not detected with less than 30% random access traffic. Thus the re-synchronization is still stable although random access messages are sent in the channel. In summary, we can extend the answer for research question two with *we see that Scheduled always outperforms random access by means of collision probability if used in coexistence.*

#### 5.1.5. Sub-Scenario Study

As a next step, it is determined whether the different sub-scenarios or the SF-based message recovery described in Section 4.3.3 can significantly improve the collision probability. Table 5 lists the decrease in collision probability for different scenarios compared to scenario S1.1 as baseline. Each line in the table shows one scenario. SF suggests message recovery because of higher SF. The columns present the collision probability decrease, for *all*, *random access*, *Scheduled*, and *systematic collisions*. The last column shows the result for the synchronization message losses. It is shown that all scenarios improve compared to the baseline scenario S1.1. The largest improvement is visible for the systematic collisions. This is for two reasons: first, for S1.2 and S1.3, less messages per hour must be transmitted and thus, it is less likely that the clock of one device drifts out of the slot before re-synchronization is performed. Second, the SF recovery works especially good for the re-synchronization messages since they are transmitted with SF 12. Furthermore, with regard to systematic collisions, it is visible that S1.2 without recovery performs much better then S1.3. The larger slots in S1.2 increase the number of re-synchronization messages that can collide before a systematic collision occurs. Since no other collision type shows a significant difference between S1.2 and S1.3, S1.2 is always preferable.

The benefit of the SF recovery, however, is also visible for all scenarios. Especially little improvement is visible for the Scheduled and sync message losses. With S1.2 and S1.3 the probability of message collision is lower but also less Scheduled messages are sent and thus less re-synchronization messages are transmitted. Especially with larger slots, this is visible for the syncs for S1.3 with a performance impairment. Furthermore, the improvement is not only visible for all data or a high random access traffic percentage but also for a little amount of random access traffic. The collision probability reduction between S1.1 and S1.1 SF is 40.83% with 10% random access traffic and 42.97% for 1% cross-traffic. There, only 0.25% Scheduled collisions are detected for S1.1, and 0.14% for S1.1 SF.

#### 5.1.6. Re-Synchronization Study

Furthermore, when studying the re-synchronization messages, it is visible that all approaches are also stable and do not exceed duty cycle limitations. For each scenario, the clock drift is chosen randomly between 0 ppm and the set limit between 2 ppm and 150 ppm. This leads to an average clock drift of 50% of the respective maximum. Thus, only a maximum of 50% from the available gateway duty cycle is used for re-synchronization.

The results for S1.1 show a a required gateway duty cycle of 44.92% for 10 ppm clock drift and 1% random access traffic and a maximum of 49.98% for 2 ppm clock drift and 50% random access traffic. The result for S1.1 SF is a little lower since less re-synchronization messages collide. This validates Equation (Equation 11) and shows that the re-synchronization limit can be adjusted according to the average clock drift if the full gateway duty cycle can be used for re-synchronization only; however, the increase in total messages per hour creates additional traffic and thus, collisions. When exhausting the duty cycle limits of the gateway, random access traffic percentages of 30% or higher are not advisable. In contrast, for S1.2 and S1.3, as expected, a different behavior is visible. There, with 50% less Scheduled messages for 50% random access traffic, duty cycle percentages between 23.73% and 25.24% are achieved. It is detected that the duty cycle requirement goes linearly down with the number of Scheduled messages.

### 5.2. Scenario S2: Scheduled and Listen before Talk

The coexistence simulation of Scheduled and listen before talk shows similar results as the random access and Scheduled study. However, since listen before talk avoids some collisions, the overall collision probability is smaller. Since differences are small, the results are only compared to the findings for scenario S1. Please note that SFs are calculated according to Section 4.2.5 for all compared scenarios here since this influences the ToA for different payloads. This guarantees comparability among scenarios.

Figure 9 shows the collision probability for the coexistence study between Scheduled and random access traffic (black) and scheduled and listen before talk traffic (brown). The *y*-axis presents the collision probability and the *x*-axis the cross-traffic other than Scheduled. The figure shows that S2.1 (listen before talk cross-traffic) always performs better than S1.1 (random access cross-traffic) while the difference is increasing with the percentage of not Scheduled traffic. This is expected since listen before talk only avoids certain collisions and does not influence other access methodologies with additional messages or overhead. Thus, for 10% not Scheduled traffic, the median collision probability for S1.1 is 3.44%, for S2.1 it is 2.96%, and the mean collision probability is 3.46% and 2.94%. The mean collision probability difference is more than 5% for 50% not Scheduled traffic. Since other values show similar results, no further description for them is given in this section.

### 5.3. Coexistence: Scheduled, Random Access, and Listen before Talk

In general, the results show that Scheduled can only coexist with random access or listen before talk as cross-traffic in a meaningful way if the predominant form is Scheduled. The difference whether the remaining traffic is transmitted with random access or listen before talk is small but listen before talk performs better. According to the evaluation of S2, the difference in the mean collision probability if using random access or listen before talk is 0.52% for example for 90% Scheduled messages. The exact value if both are used in coexistence with Scheduled is determined by the percentage of listen before talk or random access traffic while the collision probability increases with more random access and less listen before talk traffic. Furthermore, with increasing cross-traffic percentages, the risk of systematic collisions for Scheduled is also increasing. This can lead to a loss of all messages sent in specific slots in the worst case.

## 6. Conclusions

The continuous adoption of LPWAN technologies in current sensor networks requires solutions to improve transmission quality. This is especially relevant in LoRaWAN as one of the most prominent LPWAN technology since the current use of random channel access suffers from message collisions and loss. In particular, the collision rates increase with the overall transmission rates in LoRaWAN channels.

This work shows that the number of messages is not the largest impact factor for service quality degradation by collisions but the number of not planned messages. A theoretical relationship is outworked to avoid collisions completely with Scheduled if guidelines for number of messages, device clock drifts, synchronization message time on air, and duty cycle constrains are complied. It is shown that 873 messages per hour can be transmitted without collisions if device clock drifts are below 2 ppm and only the presented Scheduled channel access approach is used. Furthermore, with this message number it is also possible to re-synchronize all devices without exceeding the gateway duty cycle.

We see that slotted ALOHA is not practical in realistic network conditions with different message sizes and SFs and listen before talk is always preferred over random access; however, the study also shows that Scheduled outperforms all other channel access approaches with regard to collision probability if it is the predominant form of channel access. Furthermore, the study shows that Slotted works independent on the message rate, message size, or the spreading factor as largest influencing factors for the ToA of different devices.

The scenario study shows that even for 50% random access traffic, Scheduled messages show less than half the collision probability than random access ones. Thus, it is a valuable approach in all networks with random access-based cross-traffic; however, it must be guaranteed that re-synchronization messages are not frequently lost to avoid systematic collisions at all cost. It is shown that message recovery approaches help to prevent this situation in nearly all cases.

## Figures and Tables

**Figure 1 sensors-22-00691-f001:**
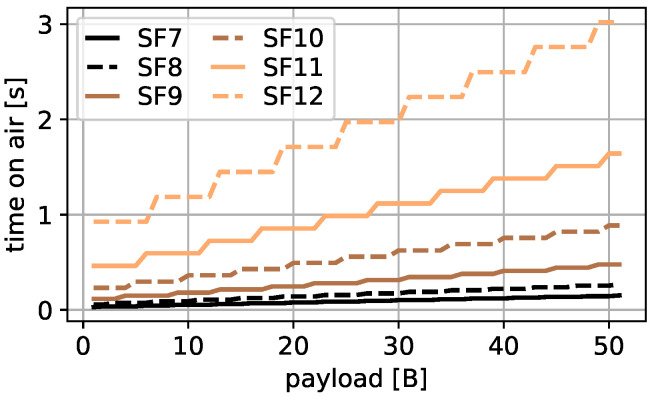
Payload and SF to ToA mapping.

**Figure 2 sensors-22-00691-f002:**
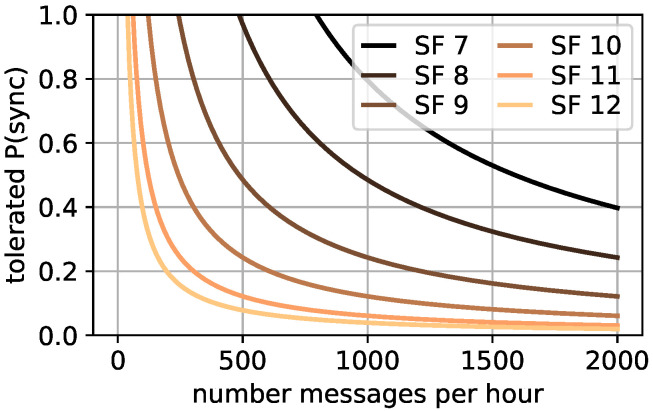
P(sync) to messages relation.

**Figure 3 sensors-22-00691-f003:**
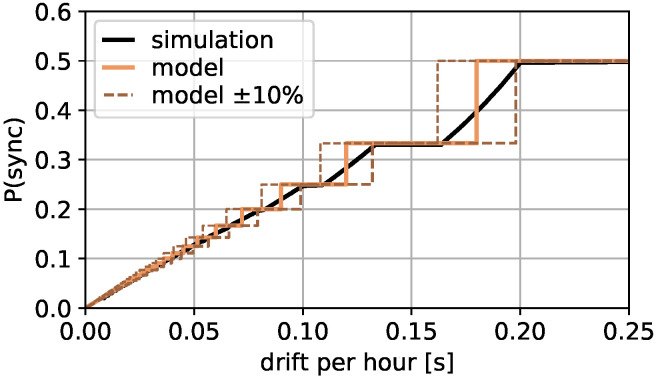
P(sync) to clock drift relation.

**Figure 4 sensors-22-00691-f004:**
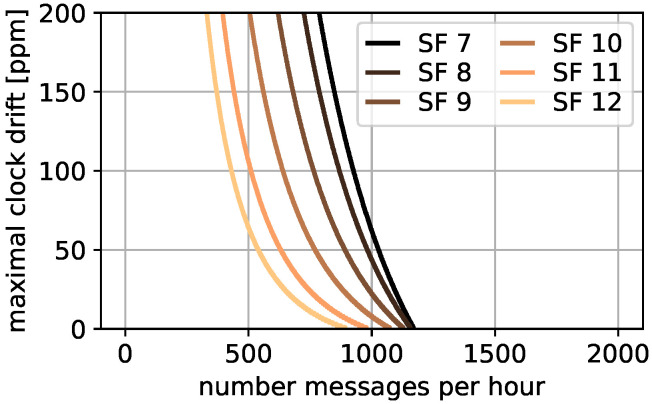
$T_{d}$ to messages relation.

**Figure 5 sensors-22-00691-f005:**
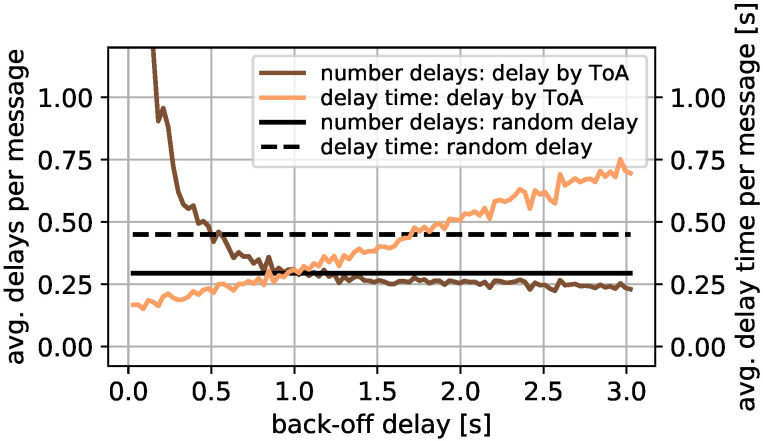
Optimal back-off study.

**Figure 6 sensors-22-00691-f006:**
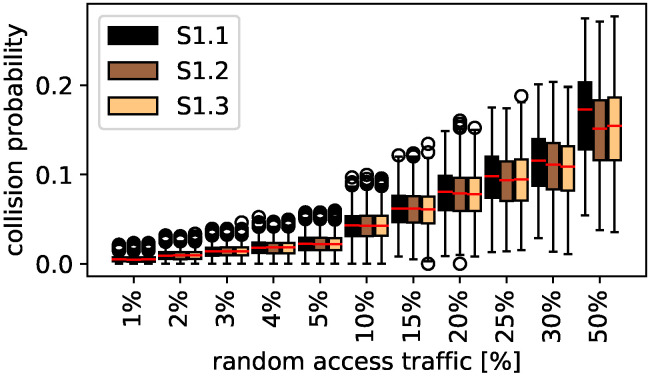
Scheduled and random access: traffic study.

**Figure 7 sensors-22-00691-f007:**
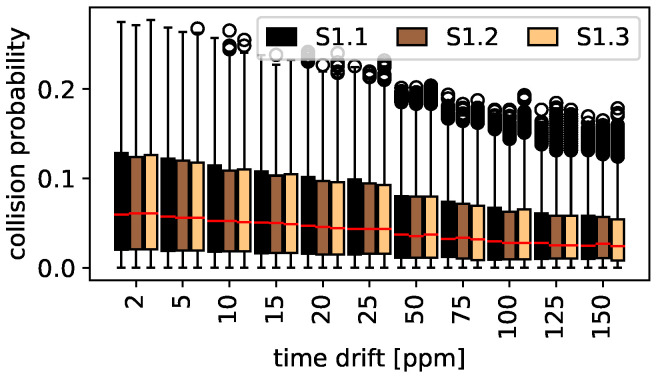
Scheduled and random access: clock drift study.

**Figure 8 sensors-22-00691-f008:**
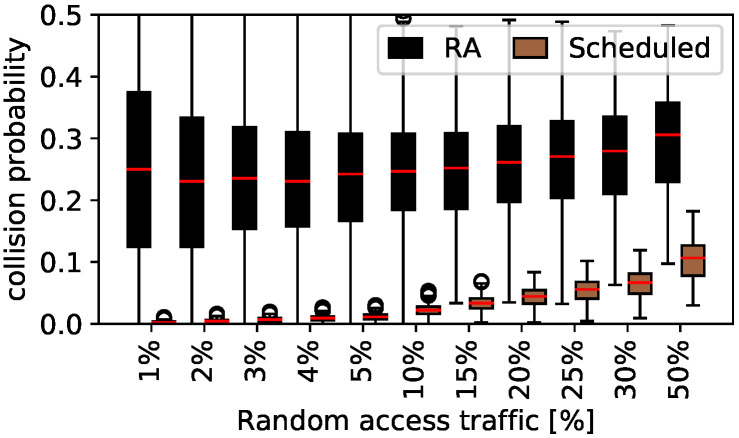
Collision by access type.

**Figure 9 sensors-22-00691-f009:**
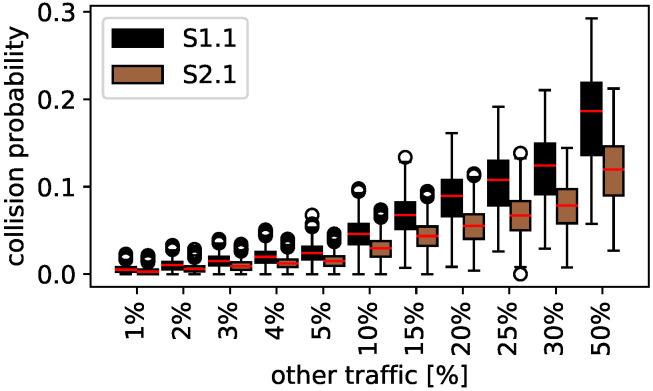
Comparison of Scheduled with random access (S1.1) and listen before talk (S2.1).

**Table 1 sensors-22-00691-t001:** Overview of important notations.

Variable	Explanation	Variable	Explanation
$T_{s}$	symbol duration	$T_{m}$	ToA for LoRa message
$T_{sync}$	ToA for re-synchronization message	$T_{d}$	clock drift
$T_{l}$	clock drift limit before re-synchronization	*r*	randomness variable
$L_{s}$	slot length	$L_{s}^{\prime }$	minimal slot length
$P_{sync}$	re-synchronization probability per message	*n*	number of messages per hour
*d*	duty cycle	$M_{x}$	message *x*
$d_{m}$	possible transmission radius around gateway	*k*	reciprocal of re-synchronization probability
$t_{e}$	transmission end time-stamp	$t_{s}$	transmission start timestamp

**Table 2 sensors-22-00691-t002:** LoRa message parameter overview.

Parameters	Variable	Value
bandwidth	BW	125 kHz
spreading factor	SF	7–12
coding rate	CR	4
payload	PL	1 B–51 B
cyclic redundancy check	CRC	1
enabled or disabled header	IH	1
low datarate optimize	DE	0
preamble length	P	8 symbols

**Table 3 sensors-22-00691-t003:** Transmission distances for different SFs.

SF	RSSI Limit [dBm]	Distance [m]	SF	RSSI Limit [dBm]	Distance [m]
7	−131	714.64	10	−140	1173.63
8	−134	843.14	11	−141	1240.12
9	−137	994.75	12	−144	1463.11

**Table 4 sensors-22-00691-t004:** Parameter settings scenario S1 and S2.

max$\left(T_{d}\right)$ [ppm]	k	n	max$\left(T_{d}\right)$ [ppm]	k	n
150	10	370	20	18	679
125	11	396	15	19	718
100	12	430	10	20	765
75	13	475	5	22	826
50	14	540	2	23	873
25	17	647			

**Table 5 sensors-22-00691-t005:** Collision probability improvement for sub-scenarios and SF-based message recovery.

	All	Random Access	Scheduled	System	Sync
**S1.1 SF**	0.428	0.476	0.415	0.999	0.830
**S1.2**	0.044	0.125	0.006	0.981	0.014
**S1.2 SF**	0.429	0.478	0.412	0.972	0.834
**S1.3**	0.044	0.126	0.004	0.472	−0.003
**S1.3 SF**	0.433	0.483	0.412	0.960	0.832

## Data Availability

Simulation scripts and evaluation scripts are publicly available at Github via https://github.com/lsinfo3/lora_without_collision_sensors, (accessed on 16 December 2021).

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
