# Peer review of "Towards LoRaWAN without Data Loss: Studying the Performance of Different Channel Access Approaches"

_sensors, 2022, doi:10.3390/s22020691_

Round 1

Reviewer 1 Report

In this paper, state of the art random channel access is compared with alternative approaches from literature by means of collision probability. Furthermore, a time scheduled channel access methodology is presented to completely avoid collisions in LoRaWAN. For this approach, an exhaustive simulation study is conducted and the performance is evaluated with random access cross-traffic. This paper investigates an interesting problem, and the structure is relative good. However, a minor revision is needed before the acceptance.

1. It seems the proposed estimation framework can works in the WAN, therefore can it be applied in the LAN scenarios. Which can make this work more meaningful.
2. For the evaluation, the dataset/algorithm choosing reasons, the detailed platform configurations and the discussion on other untested datasets should be introduced in the revised paper.
3. If the abbreviations (e.g., CSS) only appear once in the main text, authors can remove them.
4. Please add a notation table to explain all used math symbols for easy understanding.
5. Please double check whether the right template are used. Correct some formatting issues (e.g., the calibration names in Figure A2 are capitalized).
6. Some references lack the necessary information (e.g., [4]), please provide all information according to the right template.
7. Make the References more comprehensive, besides Performance of Different Channel Access, some other promising scenarios (e.g., Big data, other IoT systems) can be covered in this work. If the above related work can be discussed, it can strongly improve the research significance. For the improvement, the following papers can be considered to make the references more comprehensive.

D. S. Kapoor and A. K. Kohli, “Intelligence-based channel equalization for 4x1 SFBC-OFDM receiver,” Intelligent Automation & Soft Computing, vol. 26, no.3, pp. 439–446, 2020.

F. Toosy and M. S. Ehsan, “Statistical inference of user experience of multichannel audio on mobile phones,” Computers, Materials & Continua, vol. 65, no. 2, pp. 1253–1270, 2020.

B. L. Nguyen, E. L. Lydia, M. Elhoseny, I. V. Pustokhina, D. A. Pustokhin et al., “Privacy preserving blockchain technique to achieve secure and reliable sharing of IoT data,” Computers, Materials & Continua, vol. 65, no. 1, pp. 87–107, 2020.

S. K. Kim, M. Köppen, A. K. Bashir and Y. Jin, “Advanced ICT and IOT technologies for the fourth industrial revolution,” Intelligent Automation & Soft Computing, vol. 26, no.1, pp. 83–85, 2020.

J. Wang, Y. Yang, T. Wang, R. Sherratt, J. Zhang. Big Data Service Architecture: A Survey. Journal of Internet Technology, 2020, 21(2): 393-405

Chubo Liu, Kenli Li, Jie Liang, Keqin Li: Service Reliability in an HC: Considering From the Perspective of Scheduling With Load-Dependent Machine Reliability. IEEE Trans. Reliab. 68(2): 476-495 (2019)

J. Zhang, S. Zhong, T. Wang, H.-C. Chao, J. Wang. Blockchain-Based Systems and Applications: A Survey. Journal of Internet Technology, 2020, 21(1): 1-14

A. Chhabra, G. Singh and K. S. Kahlon, “Qos-aware energy-efficient task scheduling on hpc cloud infrastructures using swarm-intelligence meta-heuristics,” Computers, Materials & Continua, vol. 64, no. 2, pp. 813–834, 2020.

Author Response

We would like to thank the reviewer for her/his review and the valuable feedback to improve our article. Please find our comments below. We uploaded the revised article with all changes at the sensors review platform.

Reviewer comment 1: It seems the proposed estimation framework can works in the WAN, therefore can it be applied in the LAN scenarios. Which can make this work more meaningful.

Answer from authors: Indeed, the geographical size of the LoRa-based network has no influence on the performance of the approach. The influencing factors are transmission behavior of the devices (transmission rate and packet size), the number of devices, and the hidden node problem in the listen before talk scenario.

Reviewer comment 2: For the evaluation, the dataset/algorithm choosing reasons, the detailed platform configurations and the discussion on other untested datasets should be introduced in the revised paper.

Answer from the authors: The presented channel selection algorithms are selected as follows: random access since it is the state of the art approach. Listen before talk and slotted ALOHA since both approaches are most discussed and suggested approaches from literature as alternative to random access due to simplicity and performance. The simulation is conducted at different virtual machines and computers with different RAM and CPU settings. However, this has no influence on the results but only on the runtime of the tests (for example the full simulation to compare slotted and random access with 32GB RAM, Ryzen 7, 3700X 8-Core Processor takes 8-10 hours.). The simulation is only conducted once to study the practicability and compare the suggested approach with different approaches from literature. 

We did only study random traffic since this is most challenging in LoRaWAN. For more specific traffic behavior, other approaches might perform similar or even better (see slotted ALOHA in the appendix). However, in these cases, the slot length of the scheduled approach can be adjusted towards the requirements. The benefit of our approach is, that it works independent on the message rate, size, or the spreading factor. (we added two blocks for clarification, please see Section 4.3.2 and Section 6)

Reviewer comment 3: If the abbreviations (e.g., CSS) only appear once in the main text, authors can remove them.

Answer from the authors: We removed abbreviations that only appear once.

Reviewer comment 4: Please add a notation table to explain all used math symbols for easy understanding.

Answer from the authors: Thank you for this suggestion. We added Table 1 as notation table.

Reviewer comment 5: Please double check whether the right template are used. Correct some formatting issues (e.g., the calibration names in Figure A2 are capitalized).

Answer from authors: All capitalized letters in the figure axes descriptions are changed. Templates are double checked.

Reviewer comment 6: Some references lack the necessary information (e.g., [4]), please provide all information according to the right template.

Answer from authors: Missing information are added for all references.

Reviewer comment 7: Make the References more comprehensive, besides Performance of Different Channel Access, some other promising scenarios (e.g., Big data, other IoT systems) can be covered in this work. If the above related work can be discussed, it can strongly improve the research significance. [...]

Answer from authors: Thank you for this suggestion. However, we argue that especially big data or high reliable transmissions with low latency is not possible (and not intended) with LoRaWAN. Thus, no related work in this field is added to the related work. Thank you for the suggestion to add related work about other IoT technologies, in particular. We extended our related work in particular for signal decoding (please find reference 26-28 in Section 3).

Reviewer 2 Report

This paper presents an analysis of the performance of LoRAWAN under different MAC protocols: Aloha, Slotted Aloha, TDMA Slot scheduling, Listen Before Talk. The authors compard the way in which different nodes interfere among each other.

Author Response

We would like to thank the reviewer for the time spent with our article and her/his review.

Reviewer 3 Report

In the manuscript, the authors have analyzed different channel access approaches for LoRaWAN, where the collision probability is adopted as the primary evaluation metric. The manuscript is well organized and easy to follow overall. However, some concerns still need to be addressed.

  1. What is the constant 8 in Eq. (2)? Is it the preamble length?
  2. If 8 in Eq. (2) refers to the preamble length, what is the notation $n_{preamble}$ in Eq. (4)?
  3. What is the definition of notation $n_{complete}$? It seems the notation is used without any explanation.
  4. For LoRaWAN channel access, the authors have mentioned the slotted ALOHA and the alternatives such as Scheduled and listen before talk. I am not quite sure whether the authors have discussed classes A, B, and C of LoRaWAN itself.
  5. Could the authors please kindly explain how to derive Eq. (11)? It is not direct to get (11) based on Eqs (7)-(10).
  6. Does the notation “r” refer to the same physical attribute in (7) and (11)? It seems that in (7), “r” is a random variable of time in second, while a percentage in (11).
  7. In Subsection 4.3.3, should it be $t_{e_i}$ larger than $t_{s_j}$, when $M_i$ and $M_j$ overlap?
  8. Could the authors please further proofread the manuscript? There are grammar errors (e.g., Line 21) and typos (e.g., Line 392).

Author Response

We would like to thank the reviewer for her/his review and the valuable feedback to improve our article. Please find our comments below. We uploaded the revised article with all changes at the sensors review platform.

Reviewer comment 1: What is the constant 8 in Eq. (2)? Is it the preamble length?

Reviewer comment 2: If 8 in Eq. (2) refers to the preamble length, what is the notation $n_{preamble}$ in Eq. (4)?

Reviewer comment 3: What is the definition of notation $n_{complete}$? It seems the notation is used without any explanation.

Answer from authors to comment 1-3: Thank you for these comments. Indeed, the notation is not clear there. We fixed the issue. Concerning (1). The constant 8 is a constant in the LoRa message payload, see other literature (e.g. reference [6]) as reference. We changed $n_{preamble}$ to P as listed in the Table and fixed the issue with $n_{complete}$ (please see the diff in the pdf for details in Section 2.1)

Reviewer comment 4: For LoRaWAN channel access, the authors have mentioned the slotted ALOHA and the alternatives such as Scheduled and listen before talk. I am not quite sure whether the authors have discussed classes A, B, and C of LoRaWAN itself.

Answer from authors: We added an explanation to Section 2.2. Since class A has most demands and is most challenging for re-synchronization, we only addressed class A. However, all approaches that work with class A also work with class B and C since these devices are active for longer periods and can receive messages easier. 

Reviewer comment 5: Could the authors please kindly explain how to derive Eq. (11)? It is not direct to get (11) based on Eqs (7)-(10).

Answer from authors: Thank you for this comment. Indeed, the notation is not clear here and it is not clear how to derive equation 11. We changed the equation for better understanding (see details in comment for (6)). Furthermore, we added an explanation for equation 11 in particular in Section 4.2.4.

Reviewer comment 6: Does the notation “r” refer to the same physical attribute in (7) and (11)? It seems that in (7), “r” is a random variable of time in second, while a percentage in (11).

Answer from authors: Sorry for the inconsistency. It is the same random variable. We changed it in (11) and changed the order of the variables to be aligned with previous equations.

Reviewer comment 7: In Subsection 4.3.3, should it be $t_{e_i}$ larger than $t_{s_j}$, when $M_i$ and $M_j$ overlap?

Answer from authors: Thank you for this remark. We removed the mistake. 

Reviewer comment 8: Could the authors please further proofread the manuscript? There are grammar errors (e.g., Line 21) and typos (e.g., Line 392).

Answer from the authors: We did a comprehensive proofreading to improve the grammar and text quality. Please find the changes in the pdf. 

Round 2

Reviewer 3 Report

Many thanks to the efforts of the authors. I have no further comments.